# Determinants of age at first sex inequality between women and men youth in Uganda: A decomposition analysis

**Mary Luwedde** [1] *, **Quraish Sserwanja** [2], **Nehemiah Katantazi** [3]

**1** Heart for Girls Initiative Uganda, Kayunga, Uganda, **2** Programmes Department, GOAL, Khartoum, Sudan, **3** Independent Researcher, Umeå, Sweden

* luweddemary@yahoo.com

## Abstract

### Introduction

Teenage pregnancies and sexually transmitted diseases are major public health problems in Uganda. Early sexual debut is one of the main routes of these public health problems. This study aimed to identify factors that explain age at first sex inequality between men and women Ugandan youth.

### Methods

This study used secondary data from a cross-sectional Uganda demographic health survey (2016). Participants were 10 189 sexually experienced youth. Using Stata 14, intermediary analysis was done to assess the statistical association between explanatory variables and age at first sex in a multiple logistic regression analysis. Oaxaca decomposition was used to decompose factors that explain inequalities in age at first sex between men and women youth.

### Results

Intermediary results showed Islam, many household members, residing in the eastern region, and being divorced/widowed were predictors of early age at first sex. While secondary education, higher education, blue-collar jobs, and being 20 to 30 years old were protective factors against early age at first sex. Material, behavior/cultural, psychosocial, and demographic explanatory factors jointly explained a statistically significant portion of the observed gap in early age at first sex between women and men youth. More women were at a disadvantage at an early age at first sex compared to men youth. About 96.37% of this gap was explained by unequal distribution of material, behavior/cultural, psychosocial, and demographic factors between men and women youth. Relationship to household head (49%), education (16.87%), occupation (8,94%), number of household members (8.57%), using the internet (7.99%), and reading newspapers or magazines (4.39%) made a significant contribution to the explanation of early age at first sex inequality between men and women youth.

**Data Availability Statement:** The data set used is openly available upon permission from MEASURE DHS website (URL: https://www.dhsprogram.com/data/available-datasets.cfm). However, authors are

not authorized to share this data set to the public, but anyone interested in the data set can seek it with written permission from MEASURE DHS website (URL: https://www.dhsprogram.com/data/available-datasets.cfm

**Funding:** The authors received no specific funding for this work.

**Competing interests:** The authors have declared that no competing interests exist.

## Conclusions

Results showed early age at first sex inequality between women and men youth that favored men. Programs designed to address early age at first sex and related health outcomes must combat inequities in education, employment opportunities, access to sexual reproductive information through internet, and newspapers or magazines between men and women youth. They should also foster household relationships and monitor girls.

## Introduction

Early sexual debut poses a greater threat to the reproductive health of male and female youth globally. In developing countries, more than 38 million women aged 15–19 years are sexually active [1], and experience physically forced sex contrary to men [2]. Several studies have revealed that male youth are more likely to debut sexual practices at an early age [3–5] and during that time, they may not perceive the risk of having unprotected sexual intercourse with non-regular partners compared to women [6]. While early sexual initiation is more likely among female youth with low self-esteem, in men, it is done due to high self-esteem [7]. Masculinity entails men being sexual risk-takers [8]. While femininity requires women to be passive in sexual relationships and uninformed about sexual matters, this limits their ability to access information about the dangers of sex [9]. Culturally, men are rewarded for this sexual activity while women are ashamed [10]. This unequal balance of power between men and women leads to unequal access to resources, sexual and reproductive information, and services by gender. Evidence suggests that men and women who start having sex at a younger age are more likely to have multiple sexual partners in the future [11, 12], concurrent sexual partners, transactional sex [12], and are less likely to use condoms [13] than those who do not.

Early sexual debut is associated with a lowered likelihood of contraceptive use [14]. Contraceptive use in Uganda is low with only 9.4% of female adolescents using contraceptives [15]. This predisposes victims to sexually transmitted diseases (STDs) including human immunodeficiency virus (HIV), teenage pregnancies, unwanted pregnancies, abortions [14], unfavorable academic outcomes, and related complications such as obstetric fistulas [15].

Early sexual debut exposes women to unplanned pregnancies, which pushes them to look after children and alters their development plans [16]. Globally, one out of five young women are married, or in a union, before attaining the age of 18, and in developing countries, 40% of women are married before the age of 18 years [17]. Child marriages are rare for men [18]. The proportion of teenage marriages in Uganda is higher among females (3.5%) compared to male (0.2%) youth [19]. Approximately half of the Ugandan women and 40% of men aged 15–19 years have ever had sex [20]. A more recent study conducted in rural Uganda found females being less likely to have ever had sexual intercourse compared to male youth [21].

Uganda suffers serious public health issues such as teenage pregnancies and HIV/AIDS as a result of early sexual debut. Approximately 354,736 teenage pregnancies were registered in 2020 and 196,499 in the first six months of 2021 [22]. Though the whole country is affected, the Busoga region found in eastern Uganda is the most affected [19]. Additionally, due to early sex initiation, female and male youth are susceptible to HIV. The prevalence of HIV is nearly four times higher among women aged 15 to 24 compared to men of a similar age in Uganda [23].

Gender differences and inequalities influence exposure to risk factors for sexual initiation. Some studies have shown both negative and positive effects of media exposure on the initiation

of sex in youth. Odimegwu et al. reported that exposure to mass media was linked to early sexual debut for female youth but not for men [24]. Additionally, Gazendam et al. reported girls who spent more time on social media (using electronic devices) were more likely to engage in early sexual activity, an association that was less marked in boys [25].

Lack of employment and low education propels female youth to vulnerabilities of sexual exploitation for survival [26]. Community norms anticipate that women should marry and start sex before men [27]. Women in child marriages are less educated and more likely to live in rural areas [17]. They are socio-economically more vulnerable than men; particularly those in poor communities which predispose them to coercion into sexual debut and early marriage than males [28]. Furthermore, due to poverty, some parents marry off their daughters to get money for survival [17].

There is an association between family structure and early sexual debut. Female and male youth sexual activity is associated with severe family impairment in the absence of parental supervision [29]. Evidence reveals that male youth raised by men are less likely to engage in sex [30]. Furthermore, a study conducted in Uganda discovered that the absence of both parents from the household was associated with earlier sexual debut in female youth [31]. Interactions of parents with their children has a direct effect on early sexual debut. More parent-child communication (exclusively about sexual and reproductive health issues) has been linked to greater protection and lower sexual risk in both male and female youth [32].

The lack of disaggregated data by gender among youth demonstrates that gaps in specific health needs and vulnerabilities are not recognized by policymakers and program designers, posing challenges in achieving Sustainable Development Goal 5 (gender equality), and SDG 3, addressing poor health outcomes of early sex onset by gender. This paper examined age at first sex inequalities between men and women youth in Uganda using the 2016 Uganda Demographic Health Survey (UDHS) data.

## Methods

### Study context

Uganda has a population of 41 million people, of which 54% are below 18 years and over 78% are under 30 years [33]. The distribution of the population by age in Uganda is explained by the effects of excess mortality due to HIV/AIDS and a high fertility rate [33]. About 76.23% of the population stays in rural areas and agriculture is the main source of employment [34].

Although unemployment affects Uganda's entire youth population, female youth are disproportionately affected, with a rate twice that of male youth [35].

### Study design and participants

We used data from the 2016 UDHS. This UDHS was conducted between 20th June 2016 and 16th December 2016 [19]. It was a representative cross-sectional study and used validated questionnaires. The women's questionnaire was administered to women aged 15 to 49. While the men's questionnaire was administered to all men aged 15–54 in the sample of households selected for the male survey [19]. Both questionnaires collected information about household members' socio-demographic and reproductive health information. A stratified two-stage cluster sampling design was used and census enumeration areas were used as the primary sampling units [19]. The selection of households was done through equal probability systematic sampling [19].

The 2016 UDHS report has a full description of the sampling process, which can be accessed in the reference [19].

In this study, we used the Uganda national youth policy definition of youth as young people aged 12–30 [36]. While sex initiation between the ages of 18 and above was termed a delayed sex debut, sex initiation before the age of 18 was called an early sex debut. In Uganda, everyone under the age of 18 is considered a child [37]. Older people believe that abstaining from sex until the age of 18 is the greatest way to protect children [38]. A person who makes a sexual act with another who is below 18 years commits defilement and his conviction is liable to life imprisonment in Uganda.

This study only included data on sexually experienced youth aged 15–30 years. Overall, the total number of youth who participated in this survey was 15,003 of which 4,768 had never been sexually active, leading to a final sample of 10, 189 sexually experienced youth aged 15–30 (8 244 women and 1945 men). Women were more than men because all women aged 15–49 who were permanent residents of the selected households and visitors who remained in the household the night before the survey were interviewed, according to the UDHS(2016) report [19]. However, only men in a third of the selected households were sampled [19].

## Ethics approval and consent to participate

High international ethical standards are ensured during MEASURE DHS surveys and the study protocol is performed following the relevant guidelines. The UDHS 2016 survey protocol was reviewed and approved by the ICF Institutional Review Board. Written informed consent was obtained from human participants and written informed consent was also obtained from legally authorized representatives of minor participants.

## Operationalization of variables

Study variables were selected based on literature focusing on behavioral/cultural, materialist, and psychosocial theoretical perspectives of health inequality [39]. We grouped these variables into behavioral /cultural, materialist, psychosocial, and demographic factors.

## Outcome variables

The age at first sex was the health outcome variable of this study, and it was measured on a binary scale. Our outcome variable was derived from "age at first sex." A youth was considered to have an early age at first sex if he or she had his or her first sex before the age of 18, which was coded 1. While youth who had their first sex at 18 years and above were considered to have delayed age at first sex and were coded 0.

## Grouping variable

Sex of the youth was the variable used to group the study subjects. Youth were classified as either women, coded as 0, or /men, coded as 1.

## Explanatory variables

Explanatory variables were selected based on findings from literature and grouped based on behavioral/cultural, materialist, and psychosocial explanations of health inequality perspectives from the UDHS questionnaire. Explanatory variables that did not fit in the behavioral/cultural, materialist theoretical, and psychosocial perspectives were categorized as demographic variables. Explanatory variable categories were grouped into dummy variables which were used in the Oaxaca decomposition model.

**Material variables.** Education level was categorized as no education, primary, secondary, and higher level. The reference category for education level was no education. The occupation

was categorized into three categories, not working, blue-collar jobs, and white-collar jobs. Manual, household, domestic, security work, driving, and agriculture were categorized as blue-collar jobs. While sales, professional, technical, managerial, and clerical were categorized as white-collar jobs. Not working was used as the reference. The wealth quintile was calculated in the UDHS, by using principal component analysis in which scores were based on the number and kinds of consumer goods that households owned, ranging from a television to a bicycle, car, housing characteristics such as the source of drinking water, toilet facilities, and flooring materials. It was computed based on all age groups in the survey. The first quintile represents the poorest households, and the fifth quintile is the richest household. In this study, the poorest household was made the reference. The frequency of reading newspapers or magazines was categorized as, at least once a week, less than once a week, and not at all. At least once a week was the reference category.

The frequency of listening to the radio was categorized as at least once a week, and less than once a week. At least once a week was the reference category. The frequency of watching television was categorized as at least once a week, less than once a week, and not at all. At least once a week was the reference category. The frequency of using the internet last month was categorized as almost every day, less frequent, and not at all. Almost every day was the reference category.

**Behavior/Cultural variables.** Religion was categorized as, Christianity, Islam, and other religions such as traditional, Buddhism. Christianity was the reference. The number of times traveled and slept away from home in the last 12 months was coded as, 0 to once, 2 to more times, and do not know. The reference category was 0 to once.

**Psychosocial explanations.** The number of household members was grouped into, one household member, two to five household members, and more than five members. The reference category was one household member. The sex of the household head was categorized as male or female. The reference category was male.

The relationship to the household head was categorized as head, wife, daughter/son, and other relatives. The reference category was head.

**Demographic variables.** The region of residence was categorized as, central, eastern, northern, and western. The central region was the reference category. Marital status was categorized as never married, married/cohabiting, and separated/divorced/widowed. Never married was used as a reference group. The area of residence was classified as urban and rural areas. Rural area was made a reference category. The age of the respondent was grouped into 15–19, 20–24, and 25–29 years. The reference category was 15–19 years.

## Data analysis

**Descriptive statistics.** Descriptive statistics were used to get estimates of the prevalence of age at first sex by sex of youth and proportions or frequencies of material, behavior/culture, psychosocial and demographic variables across sex of youth. Sample weights were applied.

**Intermediary analysis.** The women's and men's questionnaires were utilized to create an individual dataset. The men and women datasets were apprehended together, examined for missingness and multicollinearity. Multicollinearity was assessed using the variance inflation factor (VIF) and the mean VIF was 2.95. Sample weights were used, and analysis was done using Stata 14. The statistical significance level (alpha) was <0.05. Material, behavior/cultural, psychosocial, and demographic explanatory variables were regressed with the age at first sex in a multivariable logistic regression analysis to get adjusted odds ratios and to ascertain whether, material, behavior/cultural, and demographic variables were associated with the age at first sex.

**Oaxaca decomposition analysis.** The aim of this study was addressed using Blinder-Oaxaca decomposition analysis through the Oaxaca command [40] in Stata 14. Blinder-Oaxaca decomposition analysis attributes a health gap between the two groups to the independent contributions of a group of explanatory factors [41]. Blinder-Oaxaca decomposition analysis is based on two linear regression models that are fit for each group. Non-linear (logit) blinder-Oaxaca decomposition was used because it is suitable for binary outcomes and age at first sex is a binary outcome. It explains the gap in the means (or proportion) of an outcome variable between two groups that are men and women youth in our study. In this study, the outcome was an absolute difference (proportion/prevalence difference) in age at first sex between women and men youth. The age at first sex gap (y) is then expressed as a result of differences in explanatory variables (x's), and from differences in regression coefficients. For an explanatory variable to create an important independent contribution to the age at first sex inequality, it needs to be related to the health outcome (age at first sex; indicated by the intermediary analyses) as well as unequally distributed between the comparison groups (women and men youth) indicated in the descriptive statistics.

To explain the age at first sex gap between women and men youth, all the explanatory variables stated above were added. The model provides estimates that illustrate how well the explanatory variables jointly explained the total health gap and are reported as the total explained portion (the sum of contributions of all explanatory factors) and the unexplained portion, corresponding to the fraction of the gap attributed to differences in the association to the outcome of all factors, as well as the contribution of unobserved factors.

The total contributions are expressed in both absolute terms (same as prevalence difference) and relative contributions (percentage of the absolute total health gap) with p values and confidence intervals. Individual explanatory variables' contributions to the observed health gap are also presented as absolute and relative contributions with p values, however, relative contributions are relative to the absolute explained proportion rather than the total health gap. The normalize subcommand was used to summarize the total contribution of all categories of each categorical variable which are reported in the results section. Sample weight was applied during the analysis.

## Results

### Descriptive characteristics of the study population and age at first sex

The description of study participants by sex of youth and age at first sex is presented in Tables 1 and 2. Overall, 10 189 sexually experienced youth reported their age at first sex. The prevalence of early age at first sex was higher among women (67.55%) compared to men (58.13%). While more men (41.87%) versus women youth (32.45%) reported delayed age at first sex.

Regarding material variables, overall, both women and men youth had a slight difference in wealth quintile as shown in Table 1. While there was a big difference in the level of occupation by sex of youth, more women (21.43%) vs men (2.43%) youth were unemployed, and a higher number of men had blue-collar jobs (79.48%) than women (54.54%). Generally, a higher number of men youth (14.8%, 21.26%) vs women (9.63%,13.86%) read newspapers/magazines, watched television (33.71%, 23.41%) vs women youth (22.79%, 11.15%), used internet (13.27%, 13.48%) vs women youth (5.22%, 4.89%) and more women youth (39.07%) listened to radio compared to men (26.18%). Overall, more women youth were less educated compared to men.

In terms of behavioral/cultural factors, there was a small difference in the distribution of religion between women and men youth (Table 2). In comparison to women youth, more men (45.75%) travelled frequently and slept away from home in the previous year (33.81%).

**Table 1. Weighted descriptive characteristics of the study population and age at first sex.**

| VARIABLES | WOMEN YOUTH | | MEN YOUTH | | TOTAL | PERCENT |
|---|---|---|---|---|---|---|
| | Number | % | Number | % | Number | % |
| | 8 244 | 80,91 | 1 945 | 19,09 | 10 189 | 100 |
| **OUTCOME VARIABLE: Age at first sex** | | | | | | |
| Delayed age at first sex | 2 583 | 32,45 | 805 | 41,87 | 3 388 | 34,26 |
| Early age at first sex | 5 661 | 67,55 | 1 140 | 58,13 | 6 801 | 65,74 |
| **MATERIAL VARIABLES** | | | | | | |
| **Education** | | | | | | |
| No education | 516 | 4,92 | 49 | 2,54 | 565 | 4,46 |
| Primary | 4 850 | 56,43 | 1 024 | 50,34 | 5 874 | 55,25 |
| secondary | 2 204 | 29,59 | 632 | 34,58 | 2 836 | 30,55 |
| Higher | 674 | 9,06 | 240 | 12,54 | 914 | 9,73 |
| **Occupation** *** | | | | | | |
| Un employed | 1 709 | 21,43 | 50 | 2,43 | 1 759 | 17,78 |
| Blue-collar jobs | 4 692 | 54,54 | 1 557 | 79,48 | 6 249 | 59,32 |
| White-collar jobs | 1 830 | 24,04 | 324 | 18,1 | 2 154 | 22,9 |
| **Wealth quintile** | | | | | | |
| Poorest | 1 777 | 18,12 | 375 | 16,43 | 2 152 | 17,8 |
| Poorer | 1 705 | 19,1 | 402 | 18,59 | 2 107 | 19 |
| Middle | 1 474 | 17,61 | 359 | 18,99 | 1 833 | 17,88 |
| Richer | 1 496 | 19,31 | 386 | 21,77 | 1 882 | 19,78 |
| Richest | 1 792 | 25,87 | 423 | 24,21 | 2 215 | 25,55 |
| **Frequency of reading newspapers or magazines** | | | | | | |
| At least once a week | 686 | 9,63 | 276 | 14,8 | 962 | 10,62 |
| less than once a week | 1 054 | 13,86 | 395 | 21,26 | 1 449 | 15,28 |
| Not at all | 6 504 | 76,51 | 1 274 | 63,95 | 7 778 | 74,09 |
| **Frequency of listening to the radio** | | | | | | |
| At least once a week | 3 367 | 39,07 | 529 | 26,18 | 3 896 | 36,59 |
| Less than once a week | 4 877 | 60,93 | 1 416 | 73,82 | 6 293 | 63,41 |
| **Frequency of watching television** | | | | | | |
| At least once a week | 1 655 | 22,79 | 629 | 33,71 | 2 284 | 24,89 |
| less than once a week | 884 | 11,15 | 435 | 23,41 | 1 319 | 13,51 |
| Not at all | 5 705 | 66,06 | 881 | 42,88 | 6 586 | 61,6 |
| **Frequency of using the internet last month** | | | | | | |
| Almost everyday | 388 | 5,22 | 248 | 13,27 | 636 | 6,77 |
| Less frequent | 350 | 4,89 | 252 | 13,48 | 602 | 6,54 |
| NOT at all | 7 506 | 89,89 | 1 445 | 73,25 | 8 951 | 86,68 |

% are weighted. N(unweighted) = 10 189. Missing

*** = 27.

Among the psychosocial variables, the number of household members variable had more men youth who stayed alone (11.87%) vs women (2.29%) and a higher number of women (58.87%, 38.84%) vs men (54.23%, 33.9%) who stayed with other household members. Furthermore, a greater number of men youth were household heads or sons of household heads compared to women youth. Women, on the other hand, were wives/spouses of household heads 69 times more frequently than men Table 2.

The distribution of most demographic variables (age of respondents, type of residence, and region) differed little between men and women youth. Though more men (41.91%) vs women

**Table 2. Weighted descriptive characteristics of the study population and age at first sex.**

| | FEMALE YOUTH | | MALE YOUTH | | TOTAL | PERCENT |
|---|---|---|---|---|---|---|
| VARIABLES | Number | % | Number | % | Number | % |
| | 8 244 | 80,91 | 1 945 | 19,09 | 10 189 | 100 |
| **OUTCOME VARIABLE:** Age at first sex | | | | | | |
| Delayed age at first sex | 2 583 | 32,45 | 805 | 41,87 | 3 388 | 34,26 |
| Early age at first sex | 5 661 | 67,55 | 1 140 | 58,13 | 6 801 | 65,74 |
| **BEHAVIOR/CULTURAL VARIABLES** | | | | | | |
| **Religion** | | | | | | |
| Christianity | 7 077 | 84,58 | 1 640 | 83,06 | 8 717 | 84,29 |
| Islam | 1 060 | 14,3 | 276 | 15,66 | 1 336 | 14,56 |
| Other religion | 107 | 1,12 | 29 | 1,28 | 136 | 1,15 |
| **Times traveled and slept away from home in the last 12 months** | | | | | | |
| 0 to once | 5 457 | 65,81 | 986 | 51,53 | 6 443 | 63,06 |
| 2 to more travels | 2 754 | 33,81 | 914 | 45,75 | 3 668 | 36,11 |
| Do not know | 33 | 0,38 | 45 | 2,72 | 78 | 0,83 |
| **PSYCHOSOCIAL VARIABLES** | | | | | | |
| **Number of household members** | | | | | | |
| one | 193 | 2,29 | 206 | 11,87 | 399 | 4,14 |
| 02 to 5 members | 4 814 | 58,87 | 1 059 | 54,23 | 5 873 | 57,97 |
| More than 5 members | 3 237 | 38,84 | 680 | 33,9 | 3 917 | 37,89 |
| **Sex of household head** | | | | | | |
| Male | 6 044 | 73,19 | 1 619 | 83,67 | 7 663 | 75,21 |
| female | 2 200 | 26,81 | 326 | 16,33 | 2 526 | 24,79 |
| **Relationship to the household head** | | | | | | |
| Head | 1 073 | 12,86 | 1 093 | 57,62 | 2 166 | 21,48 |
| Wife | 4 600 | 56,01 | 18 | 0,81 | 4 618 | 45,38 |
| Daughter/son | 1 265 | 15,1 | 487 | 23,86 | 1 752 | 16,78 |
| Other relatives | 1 306 | 16,04 | 347 | 17,71 | 1 653 | 16,36 |
| **DEMOGRAPHIC VARIABLES** | | | | | | |
| **Regions** | | | | | | |
| Central | 1 979 | 30,28 | 482 | 31,23 | 2 461 | 30,47 |
| Eastern | 2 339 | 27,3 | 532 | 25,83 | 2 871 | 27,01 |
| Northern | 1 855 | 18,37 | 426 | 18,13 | 2 281 | 18,32 |
| Western | 2 071 | 24,05 | 505 | 24,81 | 2 576 | 24,19 |
| **Residence** | | | | | | |
| Rural | 1 964 | 27,53 | 467 | 27,23 | 2 431 | 27,47 |
| Urban | 6 280 | 72,47 | 1 478,00 | 72,77 | 7 758 | 72,53 |
| **Age of respondent** | | | | | | |
| 15–19 | 1 655 | 19,89 | 367 | 19,1 | 2 022 | 19,74 |
| 20–24 | 3 193 | 38,97 | 715 | 36,55 | 3 908 | 38,51 |
| 25–30 | 3 396 | 41,14 | 863 | 44,34 | 4 259 | 41,75 |
| **Marital status** | | | | | | |
| Never married | 1 411 | 17,27 | 791 | 41,91 | 2 202 | 22,01 |
| Married/Living together | 6 100 | 73,46 | 1 053,00 | 52,94 | 7 153 | 69,51 |
| Separated/Divorced/Widowed | 733 | 9,27 | 101 | 5,15 | 834 | 8,47 |

% are weighted. N(unweighted) = 10189

youth (17.27%) were never married, a higher proportion of women (73.46%) vs men (52.94%) were married/living together, as shown in Table 2.

## Factors related to early age at first sex

Among all variables, secondary education, higher level of education, blue-collar jobs, white-collar jobs, Islam, staying in a household with more than five members, being related to the household head, residing in the eastern region, being 20–24 years old, 25–30 years old and being separated/divorced/widowed were statistically associated with having first sex at an early age ($P$ <0 .05) in a multivariable logistic regression analysis as shown in Tables 3 and 4. Youth with secondary education (AOR = 0.49, CI = 0.38–0.63) and higher education (AOR = 0.24, CI = 0.18–0.33) were less likely to have first sex at an early age compared to the uneducated. Similarly, youth with blue-collar jobs (AOR = 0.81, CI = 0.69–0.94), white-collar jobs (AOR = 0.81, CI = 0.68–0.97), daughters/sons (AOR = 0.70, CI = 0.56–0.89) and other relatives (AOR = 0.77, CI = 0.62–0.95) of the household heads were less likely to have first sex at an early age. Correspondingly, youth from large households of more than five members (AOR = 1.8, CI = 1.31–2.52), Islamic religion (AOR = 1.32, CI = 1.12–1.55), Eastern Uganda (AOR = 1.39, CI = 1.18–1.64), separated/divorced/widowed youth (AOR = 1.79, CI = 1.41–2.27) and being a wife of household head (AOR = 1.43, CI = 1.22–1.68) were more likely to have first sex at an early age. Whereas youth aged 20–24 (AOR = 0.21, CI = 0.17–0.25) and 25 to 30 (AOR = 0.19, CI = 0.15–0.23) were less likely to have first sex at an early age, as shown in Table 3.

**Table 3. Material factors related to early age at first sex.**

| Age at first sex | Adjusted odds ratio | P>z | [95% CI Interval] |
|---|---|---|---|
| **MATERIAL VARIABLES** | | | |
| Education (ref: no education) | | | |
| Primary | 0.93 | 0.502 | 0.73–1.16 |
| Secondary | 0.49 | **0.000** | 0.38–0.63 |
| Higher | 0.24 | **0.000** | 0.18–0.33 |
| occupation(ref: unemployed) | | | |
| Blue-collar jobs | 0.81 | **0.005** | 0.69–0.94 |
| White-collar jobs | 0.81 | **0.023** | 0.68–0.97 |
| Wealth quintile(ref: poorest) | | | |
| Poorer | 0.98 | 0.771 | 0.83–1.15 |
| Middle | 0.91 | 0.308 | 0.76–1.09 |
| Richer | 0.89 | 0.235 | 0.74–1.08 |
| Richest | 0.85 | 0.163 | 0.67–1.07 |
| Frequency of reading newspaper or magazines(ref: at least once a week) | | | |
| Less than once a week | 1.04 | 0.732 | 0.84–1.29 |
| Not at all | 1.21 | 0.066 | 0.99–1.47 |
| Frequency of listening to radio" (ref: At least once a week) | | | |
| Less than once a week | 1.02 | 0.772 | 0.91–1.14 |
| Frequency of watching television" (ref: AT least once a week) | | | |
| Less than once a week | 1.03 | 0.765 | 0.85–1.25 |
| Not at all | 0.94 | 0.431 | 0.79–1.11 |
| Frequency of using internet last month (ref: Almost every day) | | | |
| Less frequent | 1.04 | 0.798 | 0.77–1.40 |
| Not at all | 1.27 | 0.07 | 0.98–1.63 |

p-values that indicated significance at 0.05 have been indicated by using bold font.

**Table 4. Behavior/cultural, psychosocial, and demographic factors related to early age at first sex.**

| Age at first sex | Adjusted odds ratio | P>z | [95% CI Interval] |
|---|---|---|---|
| BEHAVIOR/CULTURAL VARIABLES | | | |
| Religion (ref: Christianity) | | | |
| Islam | 1.32 | **0.001** | 1.12–1.55 |
| Other religion | 1.06 | 0.785 | 0.69–1.64 |
| Times travelled and slept away from home in the last 12 months (REF:0 to once) | | | |
| 2 to more travels | 1.03 | 0.638 | 0.92–1.14 |
| Do not know | 0.73 | 0.299 | 0.40–1.32 |
| PSYCHOSOCIAL VARIABLES | | | |
| Number of household members (ref: 1 member) | | | |
| 2 to 5 household members | 1.22 | 0.194 | 0.90–1.65 |
| More than 5 members | 1.82 | **0.000** | 1.31–2.52 |
| Sex of household head (ref: male) | | | |
| Female | 1.14 | 0.086 | 0.98–1.31 |
| Relationship to household head (ref: head) | | | |
| Wife | 1.43 | **0.000** | 1.22–1.68 |
| Daughter/son | 0.70 | **0.002** | 0.56–0.89 |
| Other relatives | 0.77 | **0.017** | 0.62–0.95 |
| DEMOGRAPHIC VARIABLES | | | |
| Regions(ref: Central) | | | |
| Eastern | 1.39 | **0.000** | 1.18–1.64 |
| Northern | 0.93 | 0.417 | 0.78–1.11 |
| Western | 0.89 | 0.13 | 0.76–1.04 |
| Residence(ref: Rural) | | | |
| Urban | 0.91 | 0.213 | 0.78–1.06 |
| Age of respondent(ref:15–19) | | | |
| 20–24 | 0.21 | **0.000** | 0.17–0.25 |
| 25–30 | 0.19 | **0.000** | 0.15–0.23 |
| Marital status (ref: Never married) | | | |
| Married/Living together | 0.83 | 0.063 | 0.69–1.01 |
| Separated/Divorced/Widowed | 1.79 | **0.000** | 1.41–2.27 |

p-values that indicated significance at 0.05 have been indicated by using bold font.

## Factors contributing to age at first sex inequality between male and female youth

In the upper part of Table 5, the results show the age at first sex gap or difference between women and men youth, and a considerable portion of that gap was explained by observable characteristics, which were grouped as material, behaviour/cultural, psychosocial, and demographic variables. The "explained" part is the proportion of the difference explained by material, behavior/cultural, psychosocial, and demographic variables included in the analysis. If women and men youth had the same material, behavior, psychosocial, and demographic variables, then the "explained" portion would reduce the women-men gap in age at first sex, which is the outcome variable of interest. The lower part of Table 3 shows estimates of the contributions of material, behavior, cultural, psychosocial, and demographic variables to the explained portion of the gap.

**Table 5. Factors explaining age at first sex inequality between men and women youth.**

| Blinder-Oaxaca decomposition | | | | |
|---|---|---|---|---|
| Age at first sex | Coef. | Relative contribution (%) | P>z | 95%CI |
| Women | 0.676 | | **0.000** | 0.66–0.69 |
| Men | 0.58 | | **0.000** | 0.56–0.61 |
| Female—male difference | 0.095 | | **0.000** | 0.07–0.12 |
| Explained | 0.091 | 96.37 | **0.000** | 0.06–0.12 |
| Unexplained | 0.003 | 3.63 | 0.867 | -0.04–0.04 |
| Explained contributions | | | | |
| Material variables | | | | |
| Education | 0.016 | 16.87 | **0.000** | 0.01–0.02 |
| Occupation | 0.008 | 8.94 | **0.015** | 0.002–0.02 |
| Wealth quintile | 0.0002 | 0.26 | 0.692 | -0.001–0.001 |
| Newspapers/ magazines | 0.004 | 4.39 | **0.028** | 0.001–0.008 |
| Radio Use | -0.0005 | -0.49 | 0.767 | -0.003–0.003 |
| Television use | -0.004 | -4.24 | 0.229 | -0.01–003 |
| Internet use | 0.007 | 7.99 | **0.036** | 0.001–0.014 |
| Behavior/cultural variables | | | | |
| Religion | -0.001 | -0.77 | 0.279 | -0.002–0.001 |
| On a trip in the last 12 months | 0.0009 | 0.91 | 0.682 | -0.003–0.01 |
| Regions | 0.001 | 1.53 | 0.159 | -0.0006–0.015 |
| PSYCHOSOCIAL VARIABLES | | | | |
| Number of household members | 0.008 | 8.57 | **0.015** | 0.002–0.015 |
| Sex of household head | 0.003 | 2.79 | 0.159 | -0.001–0.006 |
| Relationship to house head | 0.0465 | 49.05 | **0.000** | 0.024–0.069 |
| Demographic variables | | | | |
| Residence | 0.00005 | 0.05 | 0.853 | -0.0005–0.0006 |
| Age | 0.0035 | 3.71 | 0.355 | -0.004–0.011 |
| Marital status | -0.0027 | -2.87 | 0.566 | -0.012–0.007 |

p-values that indicated significance at 0.05 have been indicated by using bold font.

Material, behavior/cultural, psychosocial, and demographic explanatory factors jointly explained a statistically significant and considerable portion of the observed gap in age at first sex between women and men youth as 96.37%. When it comes to the number of youth who had first sex at an early age, women outnumber men. This gap (96.37%) was explained by the unequal distribution of material, behavior, cultural, psychosocial, and demographic factors between men and women youth (Table 5). As shown in Table 5, the relationship to the household head made the biggest (49%) statistically significant contribution to the explanation of age at first sex inequality between men and women youth. This may be attributed to the substantial difference in the various types of relationships with the household heads between men and women youth (Table 2). Additionally, it was statistically associated with having first sex at an early age in the intermediary analysis of multivariable logistic regression analysis, as shown in Table 4. It was followed by education (16.87%), occupation (8.937%), number of household members (8.573%), frequency of using the internet (7.988%), and frequency of reading newspapers or magazines (4.394%). While none of the behavior/cultural and demographic variables made a significant independent contribution to the explanation of age at first sex inequality between women and men youth.

## Discussion

This is the first study to be conducted in Uganda to determine factors that explain age at first sex inequality between men and women youth. Our findings confirm the presence of early age at first sex inequality between men and women youth. Results from this study showed more women youth had first sex at an early age as young as eight years old compared to men youth. In contrast to our findings, a study conducted in rural Uganda discovered that women youth were less likely compared to men youth to have ever had sexual intercourse [21]. Additionally, studies conducted in several African countries, including Uganda, found that more Ugandan men had early sexual debut compared to women [42, 43]. In our study, the age at first sex inequality is to the advantage of the men youth, which may be attributed to the strong cultural influence that positions Ugandan women in subordinate positions but also encourages male domination even in sexuality [44]. Additionally, evidence shows that more female youth are raped/forced and coerced into early sexual activities compared to male youth [4]. Furthermore, some parents marry off their daughters to get money for survival due to poverty [17].

Our study revealed that material, behavior/cultural, psychosocial, and demographic explanatory factors jointly explained a statistically significant portion of the observed gap in age at first sex between women and man youth. This gap was explained by unequal distribution of material, behavior/cultural, psychosocial, and demographic factors between women and men youth. Among all these variables, the relationship to the household head, a psychosocial variable made the biggest (49%) statistically significant contribution to the explanation of age at first sex inequality between women and men youth. From the descriptive results, more men youth were heads of households, and more men were children of household heads compared to women youth. Yet, being a child (son or daughter) of the household head was protective against early age at first sex in Table 4. In support of this finding, Bruederle et al. found male youth raised by men, which is a proxy for the male household heads in this study, had lower odds of having sex [30]. Research done in four African countries discovered that parent-child communication about sexual matters was associated with delayed sexual debut among female youth [32]. While other research, particularly among male youth, showed the opposite [45]. Evidence shows that family structure is important to female youth's sexual behavior [31]. Parental supervision and monitoring decrease the engagement of female youth in sexual activity [46]. Monitoring and supervision of male and female youth may depend on the household adults' relationship with the youth [12].

The number of household members is another psychosocial variable, which contributed significantly to the explanation of age at first sex debut inequality between women and men youth in Uganda. Generally, from the descriptive statistics, more women youth stayed with other household members compared to men youth. In the results of Table 4, staying with many household members was a predictor of early age at first sex. This may be attributed to the lack of adequate parental monitoring and supervision of female youth because of the large household size. Yet studies have found female youth deficient in parental support are at particular risk of sexual risk-taking [30]. There is limited evidence on the relationship between the number of household members and age at first sex between men and women youth. We suggest conducting more research in Uganda to determine the relationship between household members and youths' (both male and female) first sexual experiences. Programs aimed at reducing youth sexual risk behaviors, particularly early sexual debut, must consider the diverse influences of age at first sex by gender, as well as the involvement of household heads and members. Family heads and other household members should be included in prevention measures, as well as gender-specific messages and interventions that support the program's unique aims of promoting delayed age at first sex among women and men youth.

Among material factors, education, occupation, frequency of reading newspapers or magazines, and frequency of using the internet made a significant contribution to explaining the inequality of age at first sex between men and women youth. Generally, fewer women in our study were educated, employed, read newspapers or magazines, and used the internet compared to men youth. Most women were disadvantaged in having material variables, as shown in Table 1. Some of these material variables, such as education and occupation, are protective against early age at first sex, Table 3 shows variables that have a strong association with age at first sex. Several studies found that education provided significant protection against early age at first sex among women [24, 47, 48]. This unequal balance of education and employment between women and men youth leads to unequal access to resources, reproductive health information, and health services by gender, with women being disadvantaged. Lack of employment and low education propels female youth to vulnerabilities of sexual exploitation for survival [27]. When most men are engaged in economic activities, many Ugandan women are more likely to be engaged in household tasks such as cooking, collecting firewood, water, and caring for children and ailing relatives, all of which are typically unpaid. This justifies why more women youth in our study were unemployed compared to men, yet occupation was protective against early age at first sex in Table 3.

While some studies have reported that women who used media frequently were more likely to have an early sexual debut [25], in our study more male youth used newspapers/magazines and internet compared to women youth and early age at first sex inequality was in the favour of men youth. The frequency of using newspapers/magazines and the internet contributed to the explanation of age at first sex inequality between women and men youth that favored men. We can, therefore, say that sexual information that encourages the delay of sex initiation via the media, particularly newspapers/magazines, and the internet, has not targeted women youth to address their peculiarities. Furthermore, our government's strategies for disseminating sexual knowledge through mass media, particularly newspapers/magazines and the internet, rarely address differences between women and men youth characteristics. Interventions that do not promote equal access to media, education, and employment prospects for both men and women youth expand the gender gap in age at first sex. Targeted intervention strategies should be designed to promote delay in age at first sex across gender. They should address variations in educational levels, occupation status, and access to sex education by gender. For example, policymakers must empower girls with income-generating skills, adequate information about their bodies, reproductive processes, and the advantages of delaying sexual initiation. The government of Uganda should also make the media affordable and accessible to all youth, to positively empower them with sex and reproductive health information in the early stages of their development before their sexual debut for improved sexual health. Contrary to our findings, research conducted in other settings shows that male youth who had frequent media exposure were significantly more likely to engage in early sexual debut than those who had no media exposure [49].

## Implications of the study

Inequality in early age at first sex between women and men youth favors men. To combat this challenge, gender issues need to be considered. Since the relationship to the household head contributed most to the explanation of the early age at first sex inequality and the number of household members made a significant contribution, household heads and members should educate girls about sex and monitor them during their growth. They should form bonds with all children, regardless of gender, and encourage them to adopt healthy habits, such as delaying sexual debut, at all phases of their development.

### Methodological considerations

**Strength.**   This study used data from a nationally representative standard survey, and it had a large sample size.

### Limitations

Our study used secondary data, hence limiting us to variables that were collected during the survey. We had more women compared to men youth, and it did not have important indicators of peer influence, parents' education, parental supervision, the social-economic status of the parents, whether sexual debut was forced or consented to, and it did not capture cultural as well as gender norms, which are particularly important for explaining gender inequalities in sexual debut. Furthermore, sexual behavior is a sensitive topic, and interviewing youth on sexuality may have biases that may affect the answers given. We were also unable to determine the causality or direction of relationships because we used data from cross-sectional data.

### Conclusion

Our findings demonstrate early age at first sex inequality between men and women youth in Uganda. Policies and programs designed to address early sexual debut and related health outcomes such as teenage pregnancies and HIV in youth must combat inequities in education, occupation, frequency of using the internet, and reading newspapers or magazines between men and women youth. Additionally, they should also incorporate dimensions of household heads in addition to other household members and gender-specific messages to strengthen the program's specific goals of promoting delayed sexual debut among women and men youth. Furthermore, they should encourage household heads and household members to always bond with girls and empower them with sexual reproductive information and monitor them. This study can be followed up with a qualitative study to gain a deeper understanding of this phenomenon.

### Acknowledgments

We thank the DHS program for making the data available for this study.

### Author Contributions

**Conceptualization:** Mary Luwedde.

**Formal analysis:** Mary Luwedde.

**Writing – original draft:** Mary Luwedde.

**Writing – review & editing:** Mary Luwedde, Quraish Sserwanja, Nehemiah Katantazi.

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
