## [Decision Letter · Decision Letter 0]

28 Apr 2022

PGPH-D-22-00175

Determinants of gender inequality in sexual debut among youth in Uganda:  a decomposition analysis

Dear Dr. LUWEDDE,

Thank you for submitting your manuscript to PLOS Global Public Health. After careful consideration, we feel that it has merit but does not fully meet PLOS Global Public Health’s publication criteria as it currently stands. Therefore, we invite you to submit a revised version of the manuscript that addresses the points raised during the review process.

Please submit your revised manuscript by . If you will need more time than this to complete your revisions, please reply to this message or contact the journal office at globalpubhealth@plos.org. Please include the following items when submitting your revised manuscript:

We look forward to receiving your revised manuscript.

Kind regards,

Muthusamy Sivakami

Academic Editor

Journal Requirements:

1. Please update the completed 'Competing Interests' statement. Please declare all competing interests beginning with the statement “I have read the journal's policy and the authors of this manuscript have the following competing interests:”.

2. Please amend your Financial Disclosure statement. If you did not receive any funding for this study, please simply state: “The authors received no specific funding for this work.”

Additional Editor Comments (if provided):

Dear Authors,

Thank you for submitting your paper. Our reviewers have given valuable feedback on your paper. I invite you to revise the paper based on their comments.

All comments from the reviewers are important. I want to emphasis that the paper needs to be reshaped to reflect the gender issues. Currently, the paper seriously lacks such perspectives through out the paper.

Also, please note that the paper must follow the author guidelines. Currently, the paper does not engage with the author guidelines.

Reviewers' comments:

Reviewer's Responses to Questions

**Comments to the Author**

1. Does this manuscript meet PLOS Global Public Health’s publication criteria? Is the manuscript technically sound, and do the data support the conclusions? The manuscript must describe methodologically and ethically rigorous research with conclusions that are appropriately drawn based on the data presented.

Reviewer #1: Partly

Reviewer #2: Partly

2. Has the statistical analysis been performed appropriately and rigorously?

Reviewer #1: Yes

Reviewer #2: Yes

3. Have the authors made all data underlying the findings in their manuscript fully available (please refer to the Data Availability Statement at the start of the manuscript PDF file)?

Reviewer #1: Yes

Reviewer #2: Yes

4. Is the manuscript presented in an intelligible fashion and written in standard English?

Reviewer #1: No

Reviewer #2: No

5. Review Comments to the Author

Reviewer #1: The article needs to be edited and revised by a native speaker of English for readability and coherence. Additionally, there are excess spaces and inappropriate use of punctuation throughout the paper. English usage does not always seem standard. The chart should be reformatted so that text is not truncated or hyphenated.

I appreciated the discussion of limitations, but more attention should be paid on why early marriage of women (and men) was not considered as a potential contributory factor. "Spouse of household" being more often women than men is almost self-evident, so I'm not sure how much of a value that provides as an independent variable.

Finally, setting the outcome of interest as a binary variable (sexual debut younger than, or older than, 18 years) instead of as a continuous variable was an interesting choice. It may have been required by the methodology chosen (Oaxacan decomposition analysis) but the use of a legal rule (18 years) should be justified. Is this age also meaningful statistically and in real-life? Otherwise, the use of a legal rule could be seen as arbitrary if not justified more.

The paper is compelling, timely, and important to the literature. I would recommend publication of this paper after these concerns are addressed.

Reviewer #2: The paper is taken from what seems to be a dissertation and is not written in an appropriate IMRAD format but matching that of a dissertation. A journal article has a focus that this paper lacks.

The tables are labelled ascribing methodological attributes and not representative of the table used.

The use of the word 'gender' needs strengthening through appropriate validation of the variable equivalents in the UDHS data set.

6. PLOS authors have the option to publish the peer review history of their article (what does this mean?). If published, this will include your full peer review and any attached files.

**Do you want your identity to be public for this peer review?** For information about this choice, including consent withdrawal, please see our Privacy Policy.

Reviewer #1: **Yes: **Alexander Plum

Reviewer #2: **Yes: **Mala Ramanathan

---

## [Decision Letter · Decision Letter 1]

10 Jun 2022

PGPH-D-22-00175R1

Determinants of sexual debut inequality between male and female youth in Uganda:  a decomposition analysis

Dear Dr. LUWEDDE,

Thank you for submitting your manuscript to PLOS Global Public Health. After careful consideration, we feel that it has merit but does not fully meet PLOS Global Public Health’s publication criteria as it currently stands. Therefore, we invite you to submit a revised version of the manuscript that addresses the points raised during the review process.

Please submit your revised manuscript by . If you will need more time than this to complete your revisions, please reply to this message or contact the journal office at globalpubhealth@plos.org. Please include the following items when submitting your revised manuscript:

We look forward to receiving your revised manuscript.

Kind regards,

Muthusamy Sivakami

Academic Editor

Journal Requirements:

Additional Editor Comments (if provided):

Thank you for addressing the reviewers feedback. The paper was sent again to the original reviewer and the rviewer has flagged number of issues again which needs your diligent attention. Kindly address them urgently and resubmit the paper.

Reviewers' comments:

Reviewer's Responses to Questions

**Comments to the Author**

1. If the authors have adequately addressed your comments raised in a previous round of review and you feel that this manuscript is now acceptable for publication, you may indicate that here to bypass the “Comments to the Author” section, enter your conflict of interest statement in the “Confidential to Editor” section, and submit your "Accept" recommendation.

Reviewer #2: (No Response)

2. Does this manuscript meet PLOS Global Public Health’s publication criteria? Is the manuscript technically sound, and do the data support the conclusions? The manuscript must describe methodologically and ethically rigorous research with conclusions that are appropriately drawn based on the data presented.

Reviewer #2: Yes

3. Has the statistical analysis been performed appropriately and rigorously?

Reviewer #2: (No Response)

4. Have the authors made all data underlying the findings in their manuscript fully available (please refer to the Data Availability Statement at the start of the manuscript PDF file)?

Reviewer #2: Yes

5. Is the manuscript presented in an intelligible fashion and written in standard English?

Reviewer #2: No

6. Review Comments to the Author

Reviewer #2: Table 1. Weighted descriptive characteristics of the study population and sexual debut

Table 2: Weighted descriptive characteristics of the study population and sexual debut

Comment: These tables report numbers of male and female youth using decimal places. This is because they have used weights from stata to determine the percentages. However, while reporting this, the practice is to report the actual ‘n’ of cases and using the weighted percentages. The authors need to correct this table by eliminating the decimal places in numbers and also verifying and reporting the actual numbers in the sample appropriately.

Table 3. Material factors related to early sexual debut

Comment: The authors continue to say that the significant findings are bolded. This is not conventional. The usual way to represent this is “p-values which indicated significance at 0.05 or below have been indicated by using bold font’ or some thing. But not the grammatically incorrect ‘bolded’. This had been mentioned in the earlier review as well.

The authors persist in using the words ‘intermediate analysis’ without labelling it as analysis to identify the variables that have a strong association with age at sexual debut. In so far as they have eliminated in the labelling of tables, this change needs to be reflected in the write up. It continues to be present in the write up – lines 479-480.

General comment:

The authors need to get their paper read by a native English speaker to ensure readability. In its present form, it does not address this issue.

7. PLOS authors have the option to publish the peer review history of their article (what does this mean?). If published, this will include your full peer review and any attached files.

**Do you want your identity to be public for this peer review?** For information about this choice, including consent withdrawal, please see our Privacy Policy.

Reviewer #2: **Yes: **Mala Ramanathan

---

## [Decision Letter · Decision Letter 2]

20 Jul 2022

PGPH-D-22-00175R2

Determinants of sexual debut inequality between male and female youth in Uganda:  a decomposition analysis

Dear Dr. LUWEDDE,

Thank you for submitting your manuscript to PLOS Global Public Health. After careful consideration, we feel that it has merit but does not fully meet PLOS Global Public Health’s publication criteria as it currently stands. Therefore, we invite you to submit a revised version of the manuscript that addresses the points raised during the review process.

Dear authors, Glad to  note the revised version. The paper was again sent back to one of the original reviewers. Kindly use conventional variable labels while describing them in the paper. These can be directly taken from the DHS report for Uganda. Such consistency  in labeling the variables are useful when the studies uses data of well know data sets. Also, our reviewer is of the opinion that the paper will enormously benefit from copy editing. May I encourage you to do one round of copy editing at your end so that we could proceed further.

Kindly get back to the paper in one week so that the paper can move to next stage.

We look forward to receiving your revised manuscript.

Kind regards,

Muthusamy Sivakami

Academic Editor

Journal Requirements:

Additional Editor Comments (if provided):

Dear authors, Glad to note the revised version. The paper was again sent back to one of the original reviewers. Kindly use conventional variable labels while describing them in the paper. These can be directly taken from the DHS report for Uganda. Such consistency in labeling the variables are useful when the studies uses data of well know data sets. Also, our reviewer is of the opinion that the paper will enormously benefit from copy editing. May I encourage you to do one round of copy editing at your end so that we could proceed further.

Kindly get back to the paper in one week  (July 26) so that the paper can move to next stage.

Reviewers' comments:

Reviewer's Responses to Questions

**Comments to the Author**

1. If the authors have adequately addressed your comments raised in a previous round of review and you feel that this manuscript is now acceptable for publication, you may indicate that here to bypass the “Comments to the Author” section, enter your conflict of interest statement in the “Confidential to Editor” section, and submit your "Accept" recommendation.

Reviewer #2: All comments have been addressed

2. Does this manuscript meet PLOS Global Public Health’s publication criteria? Is the manuscript technically sound, and do the data support the conclusions? The manuscript must describe methodologically and ethically rigorous research with conclusions that are appropriately drawn based on the data presented.

Reviewer #2: Yes

3. Has the statistical analysis been performed appropriately and rigorously?

Reviewer #2: Yes

4. Have the authors made all data underlying the findings in their manuscript fully available (please refer to the Data Availability Statement at the start of the manuscript PDF file)?

Reviewer #2: Yes

5. Is the manuscript presented in an intelligible fashion and written in standard English?

Reviewer #2: No

6. Review Comments to the Author

Reviewer #2: The authors have addressed all the comments carefully. However, while labelling the variables, it might be useful to distinguish between the variable - age at sexual debut - and the individuals. Therefore, while referring to the changes needed in policy, the linkage should be to 'age at sexual debut' and not males.

7. PLOS authors have the option to publish the peer review history of their article (what does this mean?). If published, this will include your full peer review and any attached files.

**Do you want your identity to be public for this peer review?** For information about this choice, including consent withdrawal, please see our Privacy Policy.

Reviewer #2: **Yes: **Mala Ramanathan

---

## [Editor Report · Decision Letter 3]

22 Aug 2022

Determinants of  age at first sex  inequality  between women and men youth in Uganda:  a decomposition analysis

PGPH-D-22-00175R3

Dear Dr LUWEDDE,

We are pleased to inform you that your manuscript 'Determinants of  age at first sex  inequality  between women and men youth in Uganda:  a decomposition analysis' has been provisionally accepted for publication in PLOS Global Public Health.

Best regards,

Muthusamy Sivakami

Academic Editor

Thank you for revising the paper again and I appreciate all the efforts.